# Warm temperature triggers JOX and ST2A-mediated jasmonate catabolism to promote plant growth

Tingting Zhu [1,2], Cornelia Herrfurth [3,4], Mingming Xin[5], Tatyana Savchenko [6], Ivo Feussner [3,4], Alain Goossens [1,2] & Ive De Smet [1,2 ✉]

Plants respond to warm temperature by increased elongation growth of organs to enhance cooling capacity. Phytohormones, such as auxin and brassinosteroids, regulate this growth process. However, our view on the players involved in warm temperature-mediated growth remains fragmentary. Here, we show that warm temperature leads to an increased expression of *JOXs* and *ST2A*, genes controlling jasmonate catabolism. This leads to an elevated 12HSO$_4$-JA level and consequently to a reduced level of bioactive jasmonates. Ultimately this results in more JAZ proteins, which facilitates plant growth under warm temperature conditions. Taken together, understanding the conserved role of jasmonate signalling during thermomorphogenesis contributes to ensuring food security under a changing climate.

[1] Department of Plant Biotechnology and Bioinformatics, Ghent University, Ghent, Belgium. [2] VIB Center for Plant Systems Biology, Ghent, Belgium. [3] Department of Plant Biochemistry, Albrecht-von-Haller-Institute for Plant Sciences and Goettingen Center for Molecular Biosciences (GZMB), University of Goettingen, Goettingen, Germany. [4] Goettingen Service Unit for Metabolomics and Lipidomics, Goettingen Center for Molecular Biosciences (GZMB), University of Goettingen, Goettingen, Germany. [5] State Key Laboratory for Agrobiotechnology, Key Laboratory of Crop Heterosis and Utilization (MOE), Key Laboratory of Crop Genomics and Genetic Improvement (MOA), Beijing Key Laboratory of Crop Genetic Improvement, China Agricultural University, Beijing, China. [6] Institute of Basic Biological Problems, Pushchino Scientific Center for Biological Research RAS, Pushchino, Russia. ✉email: Ive.DeSmet@psb.vib-ugent.be

Climate change impacts crop productivity and small increases in average temperatures are predicted to lead to large decreases in crop yield[1]. In plants, architectural adaptations to high ambient temperatures (referred to as thermomorphogenesis) allow improved cooling by evaporation to withstand warm temperatures[2]. Also in wheat, shoot elongation is promoted by high ambient temperatures[3] (Supplementary Fig. 1). In Arabidopsis thaliana, phytochromes, such as phytochrome B (phyB), and the basic helix-loop-helix transcription factor PHYTOCHROME INTERACTING FACTOR 4 (PIF4) are central and required regulators of warm temperature-mediated growth[2]. In addition, plant hormones, such as auxin and brassinosteroids, are critical regulators of thermomorphogenesis[2]. Notably, while jasmonic acid (JA) signaling plays a vital role in various stress responses[4], including wound response, cold stress, and heat stress, whether and how JA impacts warm temperature-mediated organ growth remains unknown.

JA biosynthesis and signaling pathways are well characterized[5]. The enzyme JASMONATE RESISTANT 1 (JAR1) conjugates JA with isoleucine to yield the bioactive version jasmonoyl-(L)-isoleucine (JA-Ile), which is perceived by a receptor complex including CORONATINE INSENSITIVE 1 (COI1) and a JASMONATE ZIM-DOMAIN (JAZ) protein. The JAZ proteins inhibit the functioning of target transcription factors, such as the MYCs, and are subject to 26S proteasome-mediated degradation following ubiquitination by the SCF$^{COI1}$ E3 ubiquitin ligase complex to modulate JA responses. To control JA signaling, the availability of bioactive JA (mainly JA-Ile) is a critical factor, and thus there are numerous metabolic reactions that convert JA to other active, inactive, or partially active compounds. For example, the three endoplasmic reticulum (ER)-localized cytochrome P450 oxidases CYP94B1, CYP94B3, and CYP94C1 generate the JA derivatives 12-hydroxy JA-Ile (12OH-JA-Ile) and 12-carboxy JA-Ile (12COOH-JA-Ile), the amidohydrolases IAR3 and ILL6 cleave JA-Ile back to free JA and in addition act on 12OH-JA-Ile, defining an indirect biosynthetic route to 12-hydroxy JA (12OH-JA), and the four JASMONATE-INDUCED OXYGENASES (JOXs)/JASMONIC ACID OXIDASES (JAOs) redundantly catalyze the oxidation of JA to 12OH-JA[6]. Subsequently, 12OH-JA can be sulfated by the sulfotransferase ST2A to form 12-hydroxy jasmonoyl sulfate (12HSO$_4$-JA)[6,7].

Here, we report that JAZ proteins are stabilized by a warm temperature that promotes hypocotyl elongation in Arabidopsis. This repression of COI1-mediated JAZ degradation is due to reduced levels of bioactive JA. Furthermore, complementary analyses in wheat indicate that the role of JA in regulating seedling's growth at warm temperature is conserved.

## Results and discussion
Our observation of COI1 phosphorylation in wheat seedlings exposed to 34 °C for 1 h (Supplementary Fig. 2) prompted us to investigate the role of JA signaling in warm temperature-mediated wheat growth. Wheat seedlings overexpressing Arabidopsis or wheat 12-OXOPHYTODIENOATE REDUCTASE 3 (OPR3), a key enzyme in JA biosynthesis[6], which display mild and variable elevated 3-oxo-2-(2-(Z)-pentenyl) cyclopentane-1-butyric acid (OPC4) and JA levels[8,9] (Supplementary Fig. 3), or wheat seedlings treated with JA methyl ester (MeJA) exposed to a growth-promoting temperature displayed a significantly shorter second leaf than their respective control plants or treatment (Fig. 1a, b and Supplementary Figs. 4, 5). Thus, JA represses warm temperature-mediated wheat seedling growth.

To further explore a possibly conserved role of JA and JA signaling in warm temperature-mediated growth, we then focused on hypocotyl growth in Arabidopsis, which is a hallmark for

thermomorphogenesis[2]. Warm temperature-triggered hypocotyl elongation was largely repressed by MeJA treatment at 28 °C in a concentration-dependent manner (Fig. 1c, d and Supplementary Fig. 6a, b). In contrast, the Arabidopsis aos mutant, which lacks JA[10], displays a slightly longer hypocotyl at 28 °C compared to wild type (Supplementary Fig. 6c, d). In agreement with previous observations[11], MeJA also leads to reduced Col-0 hypocotyl growth at 21 °C, but this is less pronounced than at 28 °C (Fig. 1d and Supplementary Fig. 6b). For further analyses, we used 30 µM MeJA, as this resulted in an obviously shorter hypocotyl at 28 °C compared to control plants at 21 °C. This effect of 30 µM MeJA at 28 °C is also reflected in the expression of EXPANSIN-LIKE A 2 (EXLA2), a positive regulator of cell elongation[12], and EXPANSIN A8 (EXPA8), encoding an auxin-induced cell wall-modifying enzyme involved in turgor-driven cell expansion[13], which is upregulated at 28 °C, but repressed by MeJA at 28 °C (Fig. 1e and Supplementary Fig. 7a, b). It should be noted that, in contrast to EXPA8, EXLA2 is not rapidly upregulated by 28 °C (Supplementary Fig. 7b). Furthermore, the expression of PIF4 targets INDOLE-3-ACETIC ACID INDUCIBLE 29 (IAA29) and YUC8[14] was (partially) reduced by MeJA at 28 °C (Fig. 1f and Supplementary Fig. 7a, b). Interestingly, the expression of HSP70, a transcriptional marker for warm temperature response[15], is not downregulated by MeJA at 28 °C (Supplementary Fig. 7a), indicating that the overall temperature perception is not affected. Taken together, these results indicate that JA represses warm temperature-mediated organ growth in distinct species.

Next, we checked if the MeJA-mediated repression of hypocotyl length at 28 °C required the core JA signaling components, such as COI1, the JAZ proteins, and the MYCs. The hypocotyl of the coi1-21 mutant[16], the myc2 myc3 myc4 mutant[17], and lines overexpressing JAZ proteins[18,19] were significantly longer than the wild type at 28 °C (Fig. 1g and Supplementary Fig. 8), further indicating that JA signaling exerts repression on growth at a warm temperature. Furthermore, coi1-21 and myc2 myc3 myc4 mutant seedlings were less sensitive to 30 µM MeJA at 28 °C (Fig. 1h, i and Supplementary Fig. 9a, b). In contrast, hypocotyl length of coi1-21, myc2 myc3 myc4, and 35S::JAZ1:GUS lines was hardly affected by 30 µM MeJA at 21 °C (Supplementary Fig. 10). In addition, lines producing truncated JAZ3 proteins lacking the C-terminal Jas domain that plays a key role in destabilizing JAZ repressors in response to increased JA levels[19], such as jai3-1 or 35S::JAZ3-Δjas-GFP, were also less sensitive to exogenous 30 µM MeJA application at 28 °C (Fig. 1g and Supplementary Fig. 9c, d). Conversely, the hypocotyl length of the 35S::JAZ3:GFP and 35S::JAZ1:GUS seedlings, overexpressing wild-type JAZ proteins, was still repressed by 30 µM MeJA at 28 °C, comparable to MeJA-treated Col-0 at 28 °C (Fig. 1g and Supplementary Fig. 9e). We furthermore observed that in the hypocotyl of the 35S::JAZ1:GUS line[18], the JAZ1:GUS levels strongly and within a few hours increased at 28 °C compared to 21 °C, while the JAZ1:GUS expression levels remained unaltered (Fig. 1j, k and Supplementary Figs. 7c, 11a). Similarly, in the hypocotyl of the 35S::Jas9-VENUS line[20], the Jas9-VENUS levels increased at 28 °C compared to 21 °C (Supplementary Fig. 12). Taken together, this indicates that JAZ proteins are stabilized in the hypocotyl at 28 °C. Notably, however, the strong JAZ1:GUS signal at 28 °C could be fully suppressed by 30 µM MeJA (Fig. 1j, k), indicating that JAZ degradation can be triggered by exogenous MeJA under warm temperature. However, the increased JAZ levels are in contrast to the 28 °C-mediated increase in COI1 levels (Supplementary Fig. 11b), which is similar to previous observations of a heat-mediated increase in COI1 levels[21]. Nonetheless, together our data indicate that JA represses thermomorphogenesis and that this requires the JA signaling machinery. Specifically, high levels of JAZ repressor proteins (in coi1 or 35S::JAZΔjas lines) or

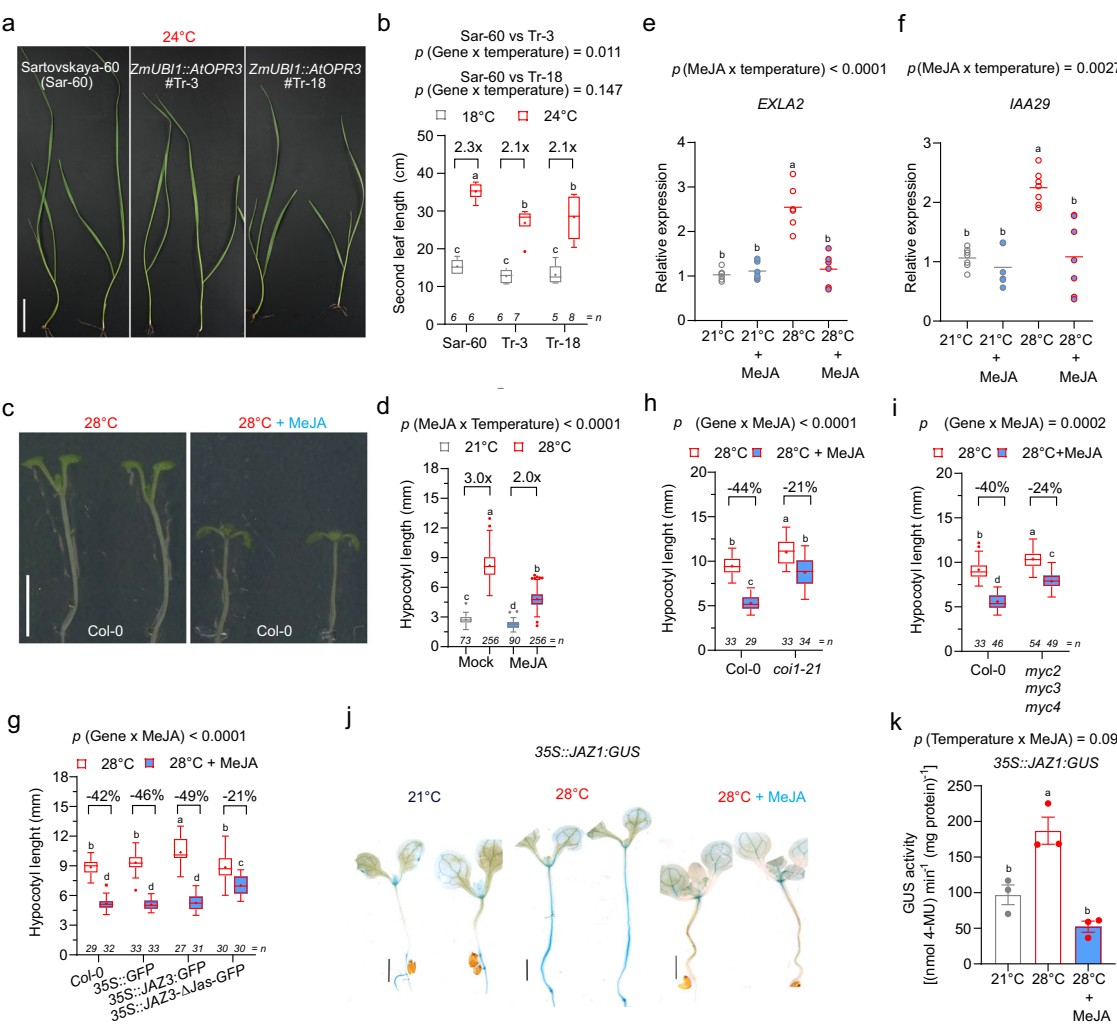

**Fig. 1 Jasmonate plays a role in thermomorphogenesis. a, b** Representative images of 10-days-old Sar-60 (parental variety) and two *AtOPR3* overexpression lines (#Tr-3 and #Tr-18) in the Sar-60 background at 24 °C (**a**) and quantification of the length of the second leaf of 10-days-old Sar-60 (parental variety) and Tr-3/-18 at 18 and 24 °C (**b**). Scale bar, 5 cm. **c, d** Representative images of 7-days-old Col-0 wild type at 28 °C in the absence or presence of 30 µM MeJA (**c**) and quantification of hypocotyl length of 7-days-old Col-0 wild type at 21 and 28 °C with or without 30 µM MeJA (data combined from multiple independent experiments). **d** Scale bar, 5 mm. **e, f** Relative expression of *EXLA2* (**e**) and *IAA29* (**f**) in 7-days-old Col-0 seedlings at 21 °C, 21 °C in the presence of 30 µM MeJA, 28 °C and 28 °C in the presence of 30 µM MeJA. Graph shows the average (line) of 5–6 biological replicates (dots). **g–i** Quantification of hypocotyl length of 7-days-old *35S::JAZ3:GFP* or *35S::JAZ3-Δjas-GFP* lines (**g**), *coi1-21* (**h**), and *myc2 myc3 myc4* mutants (**i**) and at 28 °C with or without 30 µM MeJA. **j** Representative images of GUS signal in the hypocotyl and of 7-days-old *35S::JAZ1:GUS* seedlings at 21 °C, 28 °C and 28 °C in the presence of 30 µM MeJA. Scale bar, 2 mm. **k** Quantification of GUS activity of 7-days-old *35S::JAZ1:GUS* shoots at 21, 28, and 28 °C in the presence of 30 µM MeJA. Bar diagram (**k**) shows mean of 3 biological replicates (individual dots) with standard error of the mean. Boxplots (**b, d, g–i**) show mean as "+" and show median with Tukey-based whiskers and outliers. For the quantitative figures, different letters denote significant differences ($p < 0.05$) based on two-way ANOVA with Tukey's honestly significant difference (HSD), with fold-change or % decrease between temperatures and treatment indicated (**b, d–h, i**), or on one-way ANOVA with Tukey's HSD (**k**). The number of individually measured seedlings (*n*) is indicated above the X axis (**b, d, g–i**). The exact *p*-value (or $p < 0.0001$) for the interaction (Genotype × Temperature) (**b**), (Temperature × MeJA) (**d–f**) or (Genotype × MeJA) (**g–i**) is shown at the top.

the absence of MYC-mediated response (in *myc2 myc3 myc4*) result in a long hypocotyl at 28 °C. Our data support the stabilization of JAZ proteins and reduced JA signaling at 28 °C, and furthermore suggest that this is likely, not due to an impaired perception of JA at 28 °C.

Since degradation of JAZ proteins is dependent on the availability of the bioactive JA, JA-Ile[5], we speculated that warm temperature affects the availability of bioactive JA. To test this hypothesis, we measured JA-Ile levels in both *Arabidopsis* and wheat seedlings exposed to high temperature. This indeed revealed that high temperature leads to a clear reduction in JA-Ile levels in *Arabidopsis* and to a reduction, although less

pronounced, in JA-Ile levels in wheat (Fig. 2a and Supplementary Fig. 13). However, the expression of several enzymes involved in JA biosynthesis was not affected or even up-regulated at warm temperature (Supplementary Fig. 14). Furthermore, in *Arabidopsis*, *cis*-(+)-12-oxo phytodienoic acid (OPDA) and 10-oxo *dinor*-phytodienoic acid (*dn*)-OPDA were not significantly affected and even showed a trend of increase at warm temperature like the expression of the regulatory enzymes (Supplementary Fig. 15a, b). In wheat, however, this trend is less uniform, varies for different compounds, and differs between the varieties (Supplementary Fig. 15a, c–e). Next, we, therefore, focused on the diverse catabolic processes that control the level of bioactive

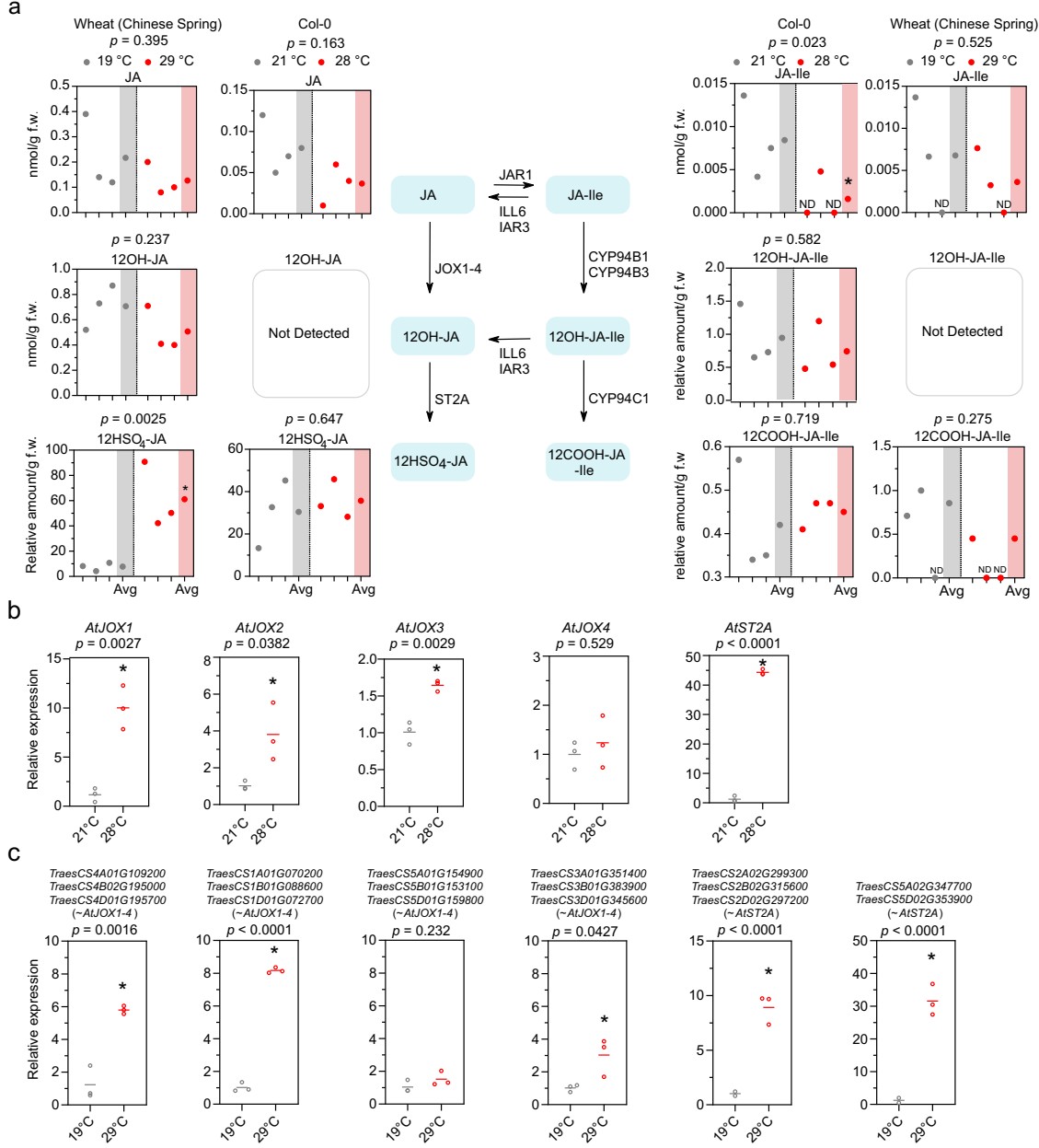

**Fig. 2 JA catabolic pathways are affected by warm temperature. a** Amounts of metabolites (JA, 12OH-JA, 12HSO$_4$-JA, JA-Ile, 12-OH-JA-Ile, 12-COOH-JA-Ile) in 7-days-old Col-0 wild type shoots at 21 °C and 28 °C or in 10-days-old Chinese Spring shoots at control temperature (19 °C) and warm growth-promoting temperature (29 °C). Samples were harvested at noon. ND, not detected (too low to be detected and considered as 0, and thus replaced with 0 when performing statistical analysis and calculating average). Dots depict 3 biological replicates (white background) and the average (Avg) (gray or red background). An * denotes significant differences ($p < 0.05$) using a Student's $t$-test with two-sample equal variance and one-tailed distribution. **b, c** Relative expression of *JOX* and *ST2A* genes in 7-days-old Col-0 wild type shoots at 21 and 28 °C collected at dawn (**b**) and of genes encoding for putative wheat *JOX* and *ST2A* orthologs in 10-days-old Chinese Spring shoots at 19 and 29 °C collected during the day (**c**). Graphs depict average (line) of 3 biological replicates (dots). An * denotes significant differences ($p < 0.05$) using a Student's $t$-test with two-sample equal variance and one-tailed distribution. The exact $p$-value (or $p < 0.0001$) is shown at the top. ns not significant.

JA-Ile, such as JOX-mediated oxidation of JA followed by ST2A-mediated sulfation, CYP94-mediated oxidation of JA-Ile and 12OH-JA-Ile, and deconjugation of JA-Ile and 12OH-JA-Ile by IAR3 and ILL6[6,7].

First, an *iar3 ill6 cyp94b1 cyp94b3 cyp94c1* pentuple mutant, which displays increased JA-Ile levels[22], displayed a slightly shorter hypocotyl at high temperature compared to the wildtype (Supplementary Fig. 16a). This further supported that reducing JA-Ile is required to promote hypocotyl growth at a warm temperature. However, while *IAR3, ILL6,* and *CYP94* expression levels

were upregulated by warm temperature in *Arabidopsis* (Supplementary Fig. 16b, c), the levels of 12OH-JA-Ile and 12COOH-JA-Ile were not higher at control versus high temperature in *Arabidopsis* or wheat (Fig. 2a and Supplementary Fig. 16d) and the *iar3 ill6* double or *cyp94b1 cyp94b3 cyp94c1* triple mutants did not display a hypocotyl phenotype at warm temperature (Supplementary Fig. 16e). This indicated that this pathway is not the main mechanism to control JA-Ile levels under warm temperature.

Second, with respect to the JOXs and ST2A-mediated catabolic pathway, 12OH-JA levels were below the detection limit in

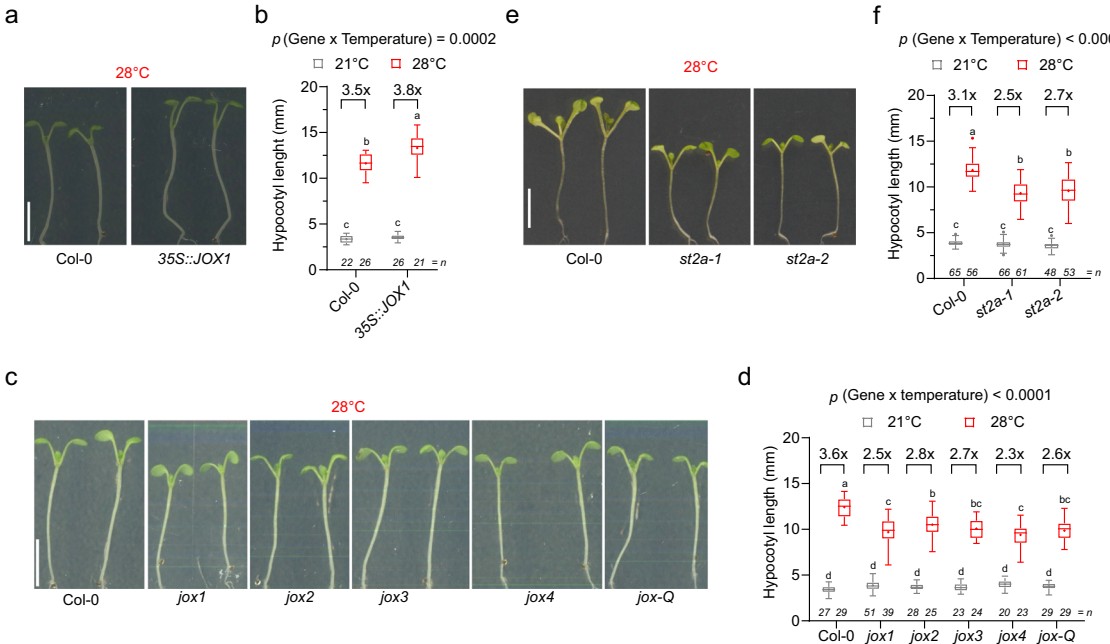

**Fig. 3 JOXs and ST2A are involved in warm temperature-mediated plant growth. a–d** Representative images at 28 °C (**a**, **c**, **e**) and quantification of hypocotyl length (**b**, **d**, **f**) of 7-days-old Col-0 and a *35S::JOX1* line (**a**, **b**), *jox-1-4* single and *jox-Q* quadruple mutant seedlings (**c**, **d**), and *st2a-1* and *st2a-2* (**e**, **f**) at 21 and 28 °C. Scale bar, 5 mm. Boxplots show mean as "+" and show median with Tukey-based whiskers and outliers, different letters denote significant differences ($p < 0.05$) based on two-way ANOVA with Tukey's HSD with fold-change between 21 °C and 28 °C being presented (**b**, **d**, **f**). The number of individually measured seedlings (*n*) is indicated above the *X* axis (**b**, **d**, **f**). The exact *p*-value (or $p < 0.0001$) for the interaction (Genotype × Temperature) (**b**, **d**, **f**) is shown at the top.

*Arabidopsis*, and 12OH-JA levels were not higher at control versus high temperature in wheat (Fig. 2a and Supplementary Fig. 17). However, while 12HSO$_4$-JA levels in *Arabidopsis* were not changed, the 12HSO$_4$-JA levels were strongly increased in wheat (Fig. 2a and Supplementary Fig. 18). In addition, expression of *JOX* and *ST2A*, genes encoding for the enzymes involved in regulating the catabolic processes leading to 12HSO$_4$-JA, was significantly upregulated at warm temperature in both wheat and *Arabidopsis*, but—at least in *Arabidopsis*—this increase in expression is time-of-day-dependent[23,24] (Fig. 2b, c and Supplementary Fig. 19). To explore whether JOX and ST2A-mediated catabolism contribute to *Arabidopsis* thermomorphogenesis, we first used a *35S::JOX1* line as a genetic tool to reduce the levels of JA and JA-Ile at high temperature[25] (Supplementary Fig. 20). We observed a longer hypocotyl of the *35S::JOX1* line at 28 °C compared to the control, while no difference was observed at 21 °C (Fig. 3a, b and Supplementary Fig. 21a). Next, to confirm the role of JOX proteins in *Arabidopsis* thermomorphogenesis, we measured the hypocotyl of *jox1*, *jox2*, *jox3*, *jox4* single mutants, and a *jox1 jox2 jox3 jox4* (*jox-Q*) quadruple mutant[25] at 21 °C and 28 °C. This revealed a significantly shorter hypocotyl and less increased growth for *jox1*, *jox2*, *jox3*, *jox4*, and *jox-Q* seedlings compared to the wildtype at 28 °C and at warm temperature relative to 21 °C, respectively (Fig. 3c, d and Supplementary Fig. 21b). The rather modest phenotype of the *jox-Q* mutants may be attributed to other catabolic pathways taking over. Indeed, we observed reduced levels of JA and JA-Ile in *jox-Q* at 28 °C, compared to the elevated levels at 21 °C (Supplementary Fig. 22). Next, we also explored if ST2A is important for thermomorphogenesis. Indeed, both *st2a-1* and *st2a-2* displayed a shorter hypocotyl at 28 °C compared to the wildtype (Fig. 3e, f and Supplementary Fig. 21c). Finally, we also observed that JOXs and ST2A are required for other thermomorphogenesis phenotypes, such as petiole length and angle[2] (Supplementary Figs. 21d and 23). The importance of

JA and JA-Ile catabolic processes at high temperature is further reflected in the increased sensitivity of *jox-Q* and *iar3 ill6 cyp94b1 cyp94b3 cyp94c1* to MeJA treatment at 28 °C (Supplementary Fig. 24). Taken together, this supports that organ growth under warm temperature conditions requires a tight control of available JA-Ile levels, mainly through JOX and ST2A-mediated catabolism. It should be noted that although *jox* and *st2a* mutants display obvious warm temperature-associated phenotypes, we did not detect 12OH-JA or a change in 12HSO$_4$-JA at warm temperature in *Arabidopsis* to support this. This may indicate that also other regulatory mechanisms, such as glycosylation of 12OH-JA[26], may be involved in the reduction of JA and JA-Ile levels at warm temperature in *Arabidopsis*.

In conclusion, warm temperature leads to an increased expression of genes controlling JA catabolism. Subsequent altered levels of bioactive JA-Ile stabilize JAZ proteins, which in turn facilitates plant growth. Similarly, the neighbor competition promotes plant growth by upregulating *ST2A* that acts to reduce the bioactive JA molecules in a phyB-dependent manner[7]. Additionally, high temperature enhances JA-dependent defense responses in tomato, resulting from increased COI1 levels and wounding-induced bioactive JA levels[21]. Our data add an additional layer of control where bioactive JA levels are tightly controlled at a warm temperature to modulate JAZ protein levels and accommodate warm temperature-mediated growth. The balance between levels of bioactive JA and the amount of COI1 determines—together with *JAZ* expression levels—JAZ protein levels and thus the response; and this seems to depend on the type of stress and even the combination of stresses. It is important to note that plants have a narrow temperature range for optimal growth and warm temperature-mediated growth promotion, and outside this range—such as extreme cold or heat—JA levels and signaling can be regulated differently leading to different responses. A broader understanding of the conserved role of jasmonate signaling when plants are exposed to low or high

temperatures contributes to ensuring food security under a changing climate.

## Methods

**Plant material and growth conditions.** All *A. thaliana* plants used in this study were in the Col-0 reference accession genetic background and referred to as wild type. The following *A. thaliana* lines were used: *aos* (ref. [10]), *coi1-21* (ref. [16]), *myc2 myc3 myc4* (ref. [17]), *jai3-1* (ref. [19]), *jox1-4* single mutants (ref. [25]), *jox-Q* (ref. [25]), *st2a-1* and *st2a-2* (ref. [7]), *iar3 ill6* (ref. [22]), *cyp94b1 cpy94b3 cpy94c1* (ref. [22]), *iar3 ill6 cyp94b1 cpy94b3 cpy94c1* (ref. [22]), *35S::JOX1* (ref. [25]), *35S::JAZ1:GUS* (ref. [18]), *35S::JAZ3:GFP* (ref. [19]), *35S::JAZ3-Δjas-GFP* (ref. [19]), *35S::Jas9-VENUS* (with *35S::H2B-RFP* background) (ref. [20]), *35S::GFP-COI1*. For phenotypic characterization, seeds were surface sterilized and vertically grown for square plates with medium containing ½ Murashige-Skoog basal salt (½ MS) (Duchefa), 1% agar, and 1% sucrose. Seeds were stratified for 3 days at 4 °C firstly and then transferred to a growth chamber at 21 °C, with continuous light (70–100 $\mu$mol m$^{-2}$ s$^{-1}$) at 50% relative humidity for germination. After germination, seeds on vertical plates were transferred to short-day conditions of 8 h light/16 h dark either at 21 °C or 28 °C (100 $\mu$mol m$^{-2}$ s$^{-1}$ photosynthetically active radiation supplied by cool-white, fluorescent tungsten tubes, Osram). Additionally, seeds for JA treatment were sown onto sterilized mesh on medium as mentioned above. After germination, the seeds with mesh were transferred to a medium as mentioned above supplemented with DMSO as mock or 30 $\mu$M MeJA (Sigma-Aldrich, CAS: 39924-52-2) for JA treatment at 21 °C or 28 °C. Plates were scanned at 7 days after germination, unless indicated otherwise. The wheat seeds used in this study were from hexaploid wheat cultivars: Chinese Spring[27], Sar-60, Tr-3 and Tr-18 (ref. [9]), KN199, and *TaOPR3*$^{OE}$ #3 (ref. [8]). The seeds were put on wet paper enclosed by plastic wrap and kept at 4 °C for 3–4 days, and then transferred to room temperature in the darkness for 1 day for germination. Seeds that germinated uniformly were selected and grown in plastic pots containing soil at the temperatures of interest under 16 h light/8 h dark (100 $\mu$mol m$^{-2}$ s$^{-1}$ photosynthetically active radiation, supplied by cool-white, fluorescent tungsten tubes, Osram) and 65–75% air humidity. In addition, for JA treatment in wheat plants, 5 ml mock (DMSO) or 200 $\mu$M MeJA solution was injected into soil grown with 1 day after germination (DAG) wheat seedlings at the temperatures of interest. The injection was applied every 3 days (three times injection in total). Phenotyping pictures were taken with a Canon digital camera. Measurement was done by using Fiji/ImageJ (https://imagej.net/Fiji). For protein extraction, 7-days-old *35S::GFP-COI1* seedlings were grown at 21 °C and 28 °C in short-day conditions on ½ MS solid medium. Or, 5-days-old *35S::Jas9-VENUS* seedlings were grown at 21 °C in short-day conditions on ½ MS solid medium, then 5-days-old seedlings were transferred to ½ MS liquid medium to adapt for 1 day at 21 °C at the same conditions. After adapting to the liquid medium, seedlings were transferred to a new liquid medium supplemented with DMSO as a mock for 20 h at 21 °C or 28 °C. Protein quantification was done by using Fiji/ImageJ (https://imagej.net/Fiji). For *GUS* expression, GUS staining and GUS activity, 5-days-old *35S::JAZ1-GUS* seedlings were grown onto sterilized mesh at 21 °C in the absence or presence of 30 $\mu$M MeJA and 28 °C in the absence or presence of 30 $\mu$M MeJA in short-day conditions on ½ MS solid medium. For short-term treatment, 5-days-old *35S::JAZ1-GUS* seedlings were grown onto mesh at 21 °C in short-day conditions on ½ MS solid medium, then 5-days-old seedlings with mesh were transferred to ½ MS solid medium as mentioned above supplemented with mock or JA treatment at 28 °C for 12 h. Or, 5-days-old Col-0 seedlings were grown onto mesh at 21 °C in short-day conditions on ½ MS solid medium, then 5-days-old seedlings with mesh were transferred to ½ MS solid medium as mentioned above supplemented with mock or JA treatment at 21 and 28 °C for 12 h.

**Gene expression analyses.** Shoot samples from 7-days-old or short-term (12 h) treated *Arabidopsis* seedlings were harvested at dawn or noon. 10-days-old wheat (Chinese Spring) seedlings were collected during the day. Three or six biological replicates were performed for each condition and each tested gene. Total RNA from *Arabidopsis* shoots or wheat seedlings was extracted and purified with the RNeasy Mini Kit (Qiagen). DNA digestion was done on columns with RNase-free DNase I (Promega). The Superscript cDNA Synthesis Kit (Quantabio) was used for cDNA synthesis from 1 $\mu$g of RNA. qRT-PCR was performed on a LightCycler 480 (Roche Diagnostics) in 384-well plates with LightCycler 480 SYBR Green I Master (Roche) according to the manufacturer's instructions. *ACTIN-RELATED PROTEIN 7* (*ARP7*) and the *TRANSLATIONAL ELONGATION FACTOR ALPHA* (*EF1α*) genes were used as *Arabidopsis* internal reference genes. *ACTIN* (GenBank locus AB181991.1) and *CELL DIVISION CONTROL PROTEIN* (GenBank locus Ta.46201) were used as wheat internal reference genes. The wheat primers were designed as follows: The wheat orthologues of the *Arabidopsis* genes were identified using gene trees available at EnsemblPlants based on the IWGSC gene models. The orthologous relationship between wheat and *Arabidopsis* genes was confirmed using reciprocal BLAST on EnsemblPlants. Each qPCR primer pair was selected on the conserved part in complete exons within homoeologues. All primers are listed in Supplementary Table 1. Analysis of relative gene expression data was performed using the $2^{-\Delta\Delta CT}$ method.

**GUS staining assay.** *35S::JAZ1-GUS* seedlings were fixed for 30 min in ice-cold 90% (v/v) acetone and rinsed with NT buffer (100 mM Tris/50 mM NaCl), incubated in X-Gluc (500 $\mu$g ml$^{-1}$ 5-bromo-4-chloro-3-indolyl-β-$_D$-glucuronide) solution (500 $\mu$l 100 mM K$_3$[Fe(CN)$_6$] + 600 $\mu$l X-Gluc + 28.8 ml NT-buffer), and incubated at 37 °C in the dark for 6-8 h. The reaction was stopped by rinsing with NT-buffer, then seedlings were mounted in 80% lactic acid on a slide and imaged.

**Quantitative GUS activity assay.** The GUS activity assay was performed according to a previously published method, with some modifications[28]. For this, 5-days-old *35S::JAZ1-GUS* shoot samples were harvested at dawn. Three biological replicates were performed for each condition. Finely ground material from 50 to 100 mg of shoots was homogenized in 500 $\mu$l of GUS extraction buffer (100 mM EDTA (pH 8.0), 0.1% SDS, 50 mM sodium phosphate (pH 7.0), 0.1% Triton X-100, adding 10 mM β-mercaptoethanol, 25 $\mu$g ml$^{-1}$ PMSF (ThermoFisher Scientific) just before use. Samples were centrifuged at 20,817×*g* at 4 °C for 10 min and the supernatant (crude extract) was transferred to new pre-cooled 1.5 ml Eppendorf tubes that were kept on ice. Next, 10 $\mu$l of crude extract was added to the 1.5 ml tubes with 37 °C prewarmed reaction mix containing 1 mM 4-MUG (Sigma-Aldrich, CAS: 6160-80-1) in GUS extraction buffer (up to 15 samples per run) within 30 seconds. After exactly 10 min of incubation at 37 °C, a 20-$\mu$l aliquot of each reaction was transferred to the first corresponding group of wells containing 180 $\mu$l stop reagent (1 M sodium carbonate) in a black 96-well cell culture plate with a flat bottom (Greiner Bio-One, CAS: 655086). After exactly 20 min of incubation at 37 °C, a 20-$\mu$l aliquot of each reaction was transferred to the second corresponding group of wells containing 180 $\mu$l stop reagent. The 4-MU (Sigma-Aldrich, CAS: 90-33-5) standard curve was prepared by diluting 1 $\mu$M 4-MU stock solution to 0, 50, 100, 200, 400, and 800 nM in stop reagent. The fluorescence of the standard curve solutions and each set of samples (10-min and 20-min samples) was measured using an excitation wavelength of 355 nm, an emission wavelength of 460 nm, and a filter wavelength of 430 nm by EnVision® 2105 multimode plate reader. A standard curve of fluorescence against concentration was plotted. The total protein concentration was determined in each sample using the Qubit™ Protein Assay Kit (ThermoFisher, USA) according to the manufacturer's instructions. The standard curve was used to calculate the amount of 4-MU per unit of time (i.e., nmol min$^{-1}$ ml$^{-1}$) for each sample. GUS activity was calculated in (nmol 4-MU) min$^{-1}$ (mg protein)$^{-1}$.

**Confocal microscopy.** Five-day-old *35S::Jas9:N7-VENUS* seedlings at 21 and 28 °C were imaged on a Leica SP8 confocal laser scanning microscope (Leica Microsystems, Germany). Scanner and detectors settings used for one experiment were optimized to avoid saturation and to maximize resolution and kept unchanged throughout the experiment. VENUS was excited using the 514 nm line (Leica SP8). VENUS fluorescence was collected from 520 to 560 nm (using the AOBS of the SP8). RFP was excited using a 561-nm laser diode (SP8). RFP fluorescence was collected from 590 to 680 nm (using the AOBS of the SP8). Images were taken using the ×40 objective, to maximize the number of nuclei being observed.

**Western blotting.** Total protein extracts were pooled from two biological replicates. Plants (*35S::GFP-COI1* and *35S::Jas9-VENUS* seedlings) were frozen in liquid nitrogen, grinded by Retsch MM400, and homogenized in 600 $\mu$l ice-cold homogenization buffer (50 mM Tris-HCl pH 7.5, 150 mM NaCl, 1% NP-40 and a Roche Complete protease inhibitor; 1 tablet/10 ml). The tubes were centrifuged for 15 min at 20,817×*g* at 4 °C and the supernatant was transferred to a new 1.5 ml tube. The protein concentration in the supernatant was determined by using the Qubit™ Protein Assay Kit (ThermoFisher, USA) according to the manufacturer's instructions. After addition of sample buffer [10×], the samples were heated for 10 min at 95 °C, centrifuged again, separated on 4–15% SDS-PAGE stain-free protein gel (Bio-Rad Laboratories, Inc., USA), and blotted on a Trans-Blot® Turbo™ Mini PVDF Transfer Packs (Bio-Rad Laboratories, Inc., USA). Membranes were blocked for 1 h at room temperature in 4% DifcoTM Skim Milk (BD Life Sciences, USA). After blocking, membranes were washed 4-5 times using PBS-T (PBS with 0.05% TWEEN® 20 (Sigma-Aldrich, USA)). For immunodetection, anti-GFP-HRP antibody (MACS, Miltenyi Biotec, catalog number: 130-091-833) at 1:5000 was used to incubate membranes for 1 h at room temperature. The proteins were detected by ChemiDoc™ MP Imaging System (Bio-Rad Laboratories, Inc., USA). After imaging, membranes were stripped using stripping buffer (1:1–1.5 ml 10% SDS: 1.5 ml 100 mM Glycine mix) for 90 s. Membranes were washed 4-5 times using dH$_2$O, followed by 3–4 times washing with PBS-T. Membranes were blocked for 1 h at room temperature and washed 4–5 times using PBS-T as described above. The membrane for proteins was immune-reacted with anti-RFP (Chromotec, catalog number:6G6) (1:2000) and mouse IgG HRP linked whole antibody (GE Healthcare, USA, catalog number:NA931) (1:10,000). The proteins were detected by a ChemiDoc™ MP Imaging System (Bio-Rad Laboratories, Inc., USA).

**Metabolite measurements.** Seven-days-old *Arabidopsis* shoot samples were harvested during the day. 10-days-old wheat seedlings were collected during the day, 2–3 wheat plant shoots were pooled as one biological replicate. Plants were frozen

in liquid nitrogen, grinded by mortar and pestle. Phytohormones were extracted with methyl-*tert*-butyl ether (MTBE), reversed phase-separated using an ACQUITY UPLC® system (Waters Corp., Milford, MA, USA) and analyzed by nanoelectrospray ionization (nanoESI) (TriVersa Nanomate®; Advion BioSciences, Ithaca, NY, USA) coupled with an AB Sciex 4000 QTRAP® tandem mass spectrometer (AB Sciex, Framingham, MA, USA) employed in scheduled multiple reactions monitoring mode[29].

**Statistical analyses and reproducibility.** Phenotypic quantification, metabolite measurement, and relative gene expression data were analyzed using either One-way or Two-way analysis of variance (ANOVA) with Tukey's honest significant difference (HSD) in GraphPad Prism 8. In some cases, data were analyzed using Students' *t*-test with two-sample equal variance and one-tailed distribution. For box plots, the lower Tukey-based whisker shows the smallest value that is greater than the lower quartile minus 1.5× interquartile range, the upper Tukey-based whisker shows the greatest value that is smaller than the upper quartile plus 1.5× interquartile range and data points outside this range (outliers) are plotted as individual dots.

**Phylogenetics.** The orthologues of the *Arabidopsis* genes in other species (*Brachypodium*, wheat, rice, maize, and barely) were identified using gene trees available at EnsemblPlants. The amino acid sequences of the orthologs were aligned by the "Align by Muscle" function and then the phylogenetic trees were reconstructed using an UPGMA statistical method implemented in MEGA5.05 and then manually edited.

## Data availability
All data are available from the corresponding author upon request. Source data are provided with this paper.

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

## Acknowledgements
We thank Clara Williams for useful discussions, Alexey V. Pigolev and Dmitry N. Miroshnichenko for generating *AtOPR3^OE* wheat lines, and Laurens Pauwels for generating *35S::GFP-COI1*. We thank G. van den Ackerveken, C. Ballaré, and S. Vanneste for providing materials. We acknowledge the Germplasm Resources Unit (GRU) at JIC for providing wheat seeds. T.Z. is supported by a grant from the Chinese Scholarship Council, I.F. acknowledges funding through the German Research Foundation (DFG, INST 186/822-1) and A.G. acknowledges funding from the Research Foundation—Flanders (FWO G008417N).

## Author contributions
I.D.S. initiated and managed the project, designed experiments, analyzed data, and wrote the manuscript; T.Z. designed and performed experiments, analyzed data, and wrote the manuscript; C.H. performed metabolite measurements, analyzed data, and commented on the manuscript; M.X. and T.S. contributed wheat lines and critically revised the manuscript; I.F. analyzed data and commented on the manuscript; A.G. designed experiments, analyzed data and commented on the manuscript. All authors discussed the results and approved the manuscript.

## Competing interests
The authors declare no competing interests.
